# Repeated games with partner choice

**Christopher Graser**[1]☯, **Takako Fujiwara-Greve**[2], **Julián García**[3], **Matthijs van Veelen**[4]☯*

**1** Dana-Farber Cancer Institute, Harvard University, Boston, Massecheusetts, United States of America,
**2** Department of Economics, Keio University, Tokyo, Japan, **3** Department of Data Science and AI, Monash University, Melbourne, Australia, **4** Department of Economics and Business, University of Amsterdam, Amsterdam, the Netherlands

☯ These authors contributed equally to this work.
* c.m.vanveelen@uva.nl

**Data Availability Statement:** This paper does not use data. The code used in our simulations is publicly available on Github: https://github.com/cjgraser/Repeated-Games-and-Partner-Choice.

## Abstract

Repetition is a classic mechanism for the evolution of cooperation. The standard way to study repeated games is to assume that there is an exogenous probability with which every interaction is repeated. If it is sufficiently likely that interactions are repeated, then reciprocity and cooperation can evolve together in repeated prisoner's dilemmas. Who individuals interact with can however also be under their control, or at least to some degree. If we change the standard model so that it allows for individuals to terminate the interaction with their current partner, and find someone else to play their prisoner's dilemmas with, then this limits the effectiveness of disciplining each other within the partnership, as one can always leave to escape punishment. The option to leave can however also be used to get away from someone who is not cooperating, which also has a disciplining effect. We find that the net effect of introducing the option to leave on cooperation is positive; with the option to leave, the average amount of cooperation that evolves in simulations is substantially higher than without. One of the reasons for this increase in cooperation is that partner choice creates endogenous phenotypic assortment. Compared to the standard models for the co-evolution of reciprocity and cooperation, and models of kin selection, our model thereby produces a better match with many forms of human cooperation in repeated settings. Individuals in our model end up interacting, not with random others that they cannot separate from, once matched, or with others that they are genetically related to, but with partners that they choose to stay with, and that are similarly dependable not to play defect as they are themselves.

## Author summary

The two mechanisms studied most in the literature on the evolution of cooperation are population structure (or kin selection), and repetition, which can allow for reciprocity to evolve. In the literature on repeated games, it is typically assumed that the matching is random and exogenous. However, not all human interactions in which there is scope for cooperation take place between individuals that have no say in who they play with, or between individuals that are genetically related. In many interactions, individuals can decide to stay with their partner, or leave and find someone else to play their repeated

**Funding:** The author(s) received no specific funding for this work.

**Competing interests:** The authors have declared that no competing interests exist.

games with. We show that if we include the option to leave in an otherwise classical setting of repeated interactions, partner choice can evolve and maintain higher levels of cooperation than reciprocity does in the standard setting, where individuals cannot leave their partner. This points to the power of partner choice.

## Introduction

In the prisoner's dilemma, repetition can stabilize cooperation. For cooperation to be stable, players need to condition their behaviour on the past actions of their interaction partner. If their partner does not cooperate, or does not cooperate enough, then reciprocal players respond with defecting, or with defecting more than they otherwise would. When faced with reciprocal partners, the self-interested thing to do can be to cooperate now in order to receive cooperation in the future. If prisoner's dilemmas are repeated, this allows for reciprocity and cooperation to evolve together [1–19].

The standard setup in models for the co-evolution of reciprocity and cooperation assumes that randomly matched individuals are tied to their partner until the repeated game ends. Who plays with whom therefore is determined exogenously. In this paper, we allow for players to end their interaction with their current partner, and look for someone else to continue playing with. Real life interactions are heterogeneous in the degree to which humans are tied to their partners. Some types of interactions allow for easy ways to change partners, others impose higher thresholds for dissolving a partnership, but all interactions find themselves somewhere on the spectrum between the standard setting, where changing partners is not possible at all, and the setting of this paper, where partners can be left, and new partners can be found, after any round of the game.

We also assume that players are not informed about their new partner's past choices. If players are informed about what their partner did in previous interactions with other players, then this could be used to enforce cooperation through norms [20], and it would allow for reputation building [21, 22], or indirect reciprocity [23, 24]. By considering a minimal setting in which no information is shared with new partners, we eliminate these possibilities. This way we isolate the role of partner choice in a minimalistic setting, without prior information (cf. [25–29]).

We analyze the evolutionary dynamics in a population playing the repeated prisoner's dilemma, and we investigate whether the option to leave undermines or facilitates the evolution of cooperation. There is a number of papers that have a setup in which players have the option to leave. There are theory papers with repeated prisoner's dilemmas, or public goods games, in which there is the option to leave [30–50], and there are empirical papers with a somewhat similar setup [51–56]. There are also theory papers in which retaliating by defecting, or by cooperating less, is not an option, but leaving is [27, 57–61].

While most of these papers do not combine a full game-theoretical analysis with studying the evolutionary dynamics, this literature does contain findings that are relevant for the dynamics. The most important one is a result that states that with the option to leave, there are no equilibria in which all players start cooperating in the first round of every new interaction ([41], see also [31, 35, 36]). The rationale for this is straightforward. Any population in which all individuals do start cooperating right from the beginning can be invaded by a mutant that takes advantage of this, by defecting and leaving after the first period. Such a mutant would get the highest possible payoff in every round, while the resident could at most earn an average payoff equal to the payoff of mutual cooperation. Without the option to leave, there are

equilibria with full cooperation, and the fact that fully cooperative equilibria do not exist if leaving is allowed for is a reflection of the downside of the option to leave, which would allow for cheaters to get away with defection and escape punishment whenever cooperation starts in the first round.

To prevent invasions by defect-and-run mutants, a simple solution could be to start every new partnership with a defection. Depending on the parameters of the game, however, one round of mutual defection may not be enough to avoid exploitation. For some combinations of the benefit-to-cost ratio and the continuation probability, starting to cooperate in round 2 may still leave the door open for a mutant that sits out one round of mutual defection, then defects on a resident that starts cooperating in round 2, and subsequently leaves in order to repeat this with its next partner. In Theorem 4 in S1 Text we specify the minimum length of this initial string of defections. This theorem is a simpler version of a result that implies that as soon as this threshold is met, equilibria that cooperate afterwards do exist [41]. The initial string of defections is sometimes also referred to as the *trust-building phase* [41–43, 47, 48]. This label may not perfectly match what we think of as trust-enhancing behaviour, but the idea is that what builds trust here is the staying, in anticipation of future mutual cooperation, and not the defecting. We will follow the existing literature in using this term.

The fact that the option to leave rules out fully cooperative equilibria, and may require multiple periods of trust-building, suggests that leaving might be bad for the evolution of cooperation. Below, we will see that this is not the case; average amounts of cooperation go up rather than down if the option to leave is added. There is a variety of reasons why the option to leave can also foster higher average cooperation rates. In Section **Relative stability of cooperative equilibria with and without leaving**, we will see that punishing by leaving creates endogenous assortment, and that this assortment can make equilibria that punish by leaving more stable than similar equilibria that punish with defection. In Section **Getting away from AllD at a higher rate** we will also see that transitions out of fully defecting equilibria happen more readily when leaving is allowed for.

## Materials and methods

### The model setup

In this paper, we restrict attention to prisoner's dilemmas with equal gains from switching [4], where the cost of cooperating instead of defecting is $c$, irrespective of whether the opponent cooperates or defects, and the benefits to the other player are $b$, again irrespective of what the opponent plays herself.

$$\begin{bmatrix} & C & D \\ C & b-c & -c \\ D & b & 0 \end{bmatrix}$$

In order to simplify the notation and analysis further, and without loss of generality, we normalize the costs to $c = 1$. This means that we can interpret the $b$ in the payoff matrix as the benefit-to-cost ratio.

Strategies are represented by finite state automata (FSAs). Fig 1 depicts an example of an FSA. The colours of the states represent the output when the FSA is in this state: red means defect; blue means cooperate; and black means that this FSA terminates the interaction. All FSAs start in the leftmost state when they begin interacting with a new partner, and the arrows indicate to which state the FSA moves in response to their partner's action. After termination, the FSA does not have to transition to any state; it restarts the interaction with its new partner

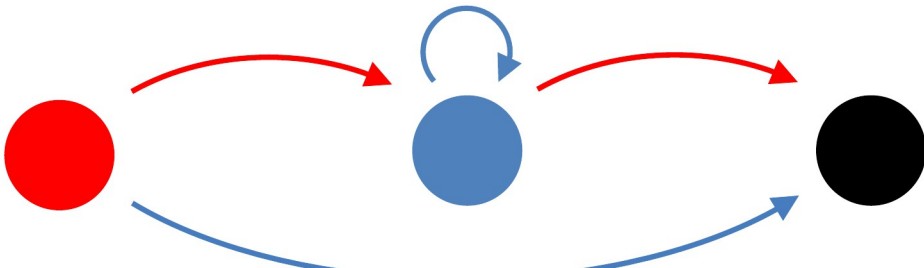

**Fig 1. Strategy $c_1$.** This finite state automaton (FSA) has three states; in state 1, the output is $D$ (red); in state 2 the output is $C$ (blue); and in state 3, the output is to leave (black). The arrows indicate to which state the strategy goes, after observing an action ($C$ or $D$) by its opponent. The FSA starts every interaction in the leftmost state.

in the leftmost state. Representing strategies as FSAs allows agent-based simulations to explore a very rich space of strategies; any thinkable strategy can be approximated arbitrarily closely by an FSA, and all FSA's can be reached by a finite sequence of mutations [17, 62].

In the model, we assume that individuals are matched to play the prisoner's dilemma. After every round, each pair is broken up exogenously with probability $1 - \delta$, where $\delta \in (0, 1)$. Pairs can also be broken up because one of the players, or both, choose to end the interaction. All broken-up pairs go to the *matching pool*, in which they are re-matched before the subsequent stage game starts. Re-matching happens uniformly at random; all pairs of individuals from the matching pool are equally likely to be formed. The matching pool is not a random draw from the population as a whole. If it would only contain individuals coming from pairs that are broken up exogenously, then the frequencies in the matching pool would match the frequencies in the population as a whole. However, the matching pool also contains individuals that broke up with their partner themselves, and individuals that are broken up with. Whether that happens is determined by the combination of strategies in the pair.

For the theoretical results, we assume an infinitely large population. This allows us to calculate payoffs for all strategies that are present in the population, assuming that the dynamics of separating and matching reach a steady state. This steady state, or short-run equilibrium, describes the shares of pairs consisting of different combinations of two strategies, and the round of the game they find themselves in. Here we will give a simple example to illustrate how that works.

If there is only one strategy present in the population, and if that strategy never leaves its partner, then all pairs consist of two individuals that both play this one strategy. In principle it is possible that all pairs are in their first round of play. As pairs are broken up randomly, however, over time, the population will converge to a state in which the ratio of pairs in their first round to those in their $n$th round is 1 to $\delta^{n-1}$. The intuition for this is that all new pairs start in round 1, while the probability for any pair making it to the $n$th round is $\delta^{n-1}$ (see Table 1).

We then calculate the payoffs by taking a weighted average over the payoffs in the different rounds, where the weight of the payoffs in the $n$th round is proportional to $\delta^{n-1}$. Without the option to leave, averaging all stage game payoffs over the population at a steady state is equivalent to the standard calculation of expected payoffs in repeated games (see S1 Text).

If the population includes strategies that may choose to leave, however, the calculation of such a steady state becomes more complicated [41, 47]. As mentioned above, if there are combinations of strategies in which one of the partners chooses to leave, then the shares of the different strategies in the matching pool and the shares of the different strategies in the

**Table 1. Shares of pairs in various rounds.**

| time \ round | 1 | 2 | 3 | 4 | $\cdots$ | n | $\cdots$ |
|---|---|---|---|---|---|---|---|
| $t = 1$ | 1 | 0 | 0 | 0 | $\cdots$ | 0 | $\cdots$ |
| $t = 2$ | $1 - \delta$ | $\delta$ | 0 | 0 | $\cdots$ | 0 | $\cdots$ |
| $t = 3$ | $1 - \delta$ | $\delta(1 - \delta)$ | $\delta^2$ | 0 | $\cdots$ | 0 | $\cdots$ |
| $t = 4$ | $1 - \delta$ | $\delta(1 - \delta)$ | $\delta^2(1 - \delta)$ | $\delta^3$ | $\cdots$ | 0 | $\cdots$ |
| $\vdots$ | $\vdots$ | $\vdots$ | $\vdots$ | $\vdots$ | | | |
| $t = n$ | $1 - \delta$ | $\delta(1 - \delta)$ | $\delta^2(1 - \delta)$ | $\delta^3(1 - \delta)$ | $\cdots$ | $\delta^{n-1}$ | |
| $t = n + 1$ | $1 - \delta$ | $\delta(1 - \delta)$ | $\delta^2(1 - \delta)$ | $\delta^3(1 - \delta)$ | $\cdots$ | $\delta^{n-1}(1 - \delta)$ | $\delta^n$ |

We start with an infinitely large population in which every pair is in the first round. The size of this population is normalized to 1. After playing the stage game, all pairs break up with probability $1 - \delta$, and stay together with probability $\delta$. For $n > 1$, the share of pairs of the population that is in round $n$ at time $t$ is $\delta$ times the share of pairs that is in round $n - 1$ at time $t - 1$. Because all pairs break up with probability $1 - \delta$, the share of newly formed pairs after $t = 1$ is always $1 - \delta$. The ratio of pairs that are in the first round of play, and pairs that are in their $n$th round of play, becomes 1 to $\delta^{n-1}$ from $t = n + 1$ onwards.

population as a whole can diverge. We describe the way of calculating the short-run equilibrium and the expected payoffs that this short-run equilibrium implies in detail in S1 Text.

## Long-run equilibrium and indirect invasions

The steady state or short-run equilibrium takes the composition of the population as given. In the long run, however, mutation and selection can change the composition of the population. A given composition of the population may or may not be a Nash equilibrium, and in order to determine if it is, we compare the payoffs of the strategies currently present in the population with each other, and with the payoffs that alternative strategies would have, if they were to enter in the population at an infinitesimal small frequency. We also use this separation of time-scales when we apply other equilibrium concepts, like evolutionary stability, neutral stability, and robustness against indirect invasions, where we also assume that the population is in short-run equilibrium in order to calculate the expected payoffs of all strategies.

For repeated prisoner's dilemmas without the option to leave, we know that there are many strategies that are neutrally stable (NSS) [6], and we know that there are no finite mixtures of strategies that are robust against indirect invasions (RAII) [11, 17, 63]. This means that in finite populations, every Nash equilibrium can be invaded indirectly; for every Nash equilibrium, there is a neutral mutant that, if it goes to fixation, opens the door for a second mutant, that then has a selective advantage. This theoretical result is matched by the fact that in simulations without the option to leave, all Nash equilibria are indeed left in due time. Moreover, for reasonably large population sizes, all of those transitions out of equilibria happen through indirect invasions [11, 17].

With the option to leave, this remains true; all finite mixtures that are Nash equilibria can be invaded indirectly (see Theorems 1 and 2 in S1 Text for a formal proof of the claim that also with the option to leave, there are no pure Nash equilibria that are RAII. The extension to finite mixtures is also discussed in S1 Text). The reason is similar to the reason without the option to leave, and is explained more easily for pure equilibria. For pure equilibria with positive amounts of cooperation, this cooperation needs to be stabilized with the threat of punishment—which can be to defect (or to defect more than the strategy would otherwise), or to leave. When a population finds itself in such an equilibrium, this punishment is not executed.

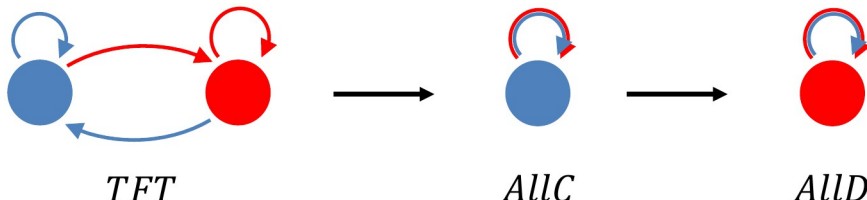

**Fig 2. An indirect invasion into Tit-for-Tat.** AllC is a neutral mutant of Tit-for-Tat; both always cooperate with copies of themselves and each other. If AllC goes to fixation by neutral drift, AllD can invade.

A mutant that has lost the capacity to punish therefore would be neutral. If random drift allows this neutral mutant to go to fixation, it would open the door for a second mutant, that takes advantage of the loss of the capacity to punish (Fig 2).

An equilibrium with defection only can also be invaded indirectly. Without the option to leave, this would require a mutant that *would* cooperate, if its partner initiates it. This is a neutral mutant, and it would open the door for a second mutant that reaps the rewards for initiating cooperation (see Fig 3). This stepping stone path requires a minimum $\delta$ for the second mutant to have a payoff advantage, and it is still there as a path out of full defection if leaving is allowed for.

With the option to leave, there is also an additional stepping stone path out of fully defecting strategies. A strategy that defects and leaves would be a neutral mutant of any fully defecting strategy. If this strategy takes over the population, it opens the door for a mutant that defects, stays, and cooperates forever after, if it finds its partner has stayed as well (see Fig 4). Importantly, this path out does *not* require a minimum $\delta$ to constitute an indirect invasion. For low $\delta$, equilibria without any cooperation therefore are less stable with the option to leave than they are without the option to leave.

For sufficiently high $b$ and $\delta$, there are stepping stone paths out of any Nash equilibrium, both with and without the option to leave. We therefore expect populations to visit a variety of equilibria, and to transition between them through indirect invasions. Which strategies are and which are not Nash equilibria, however, differs between the two settings. Tit-for-Tat, for example, is an equilibrium without the option to leave, provided that $b$ and $\delta$ are sufficiently high, while it stops being an equilibrium if leaving is allowed for. The reason why it cannot be an equilibrium with the option to leave is that it cooperates in the first round. The strategy $c_1$, as depicted in Fig 1, on the other hand, is an equilibrium when leaving is possible, provided that $b$ and $\delta$ are sufficiently high, but since it has a state in which it leaves, this obviously is not a feasible strategy if leaving is not allowed for.

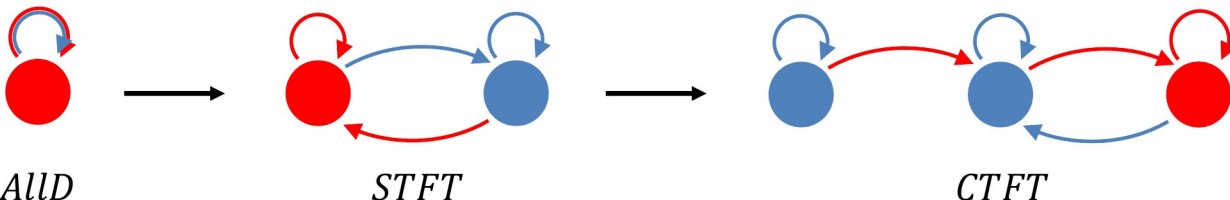

**Fig 3. An indirect invasion into AllD.** Suspicious Tit-for-Tat is a neutral mutant of AllD; both always defect with copies of themselves and each other. If Suspicious Tit-for-Tat goes to fixation by neutral drift, C-Tit-for-Tat can invade.

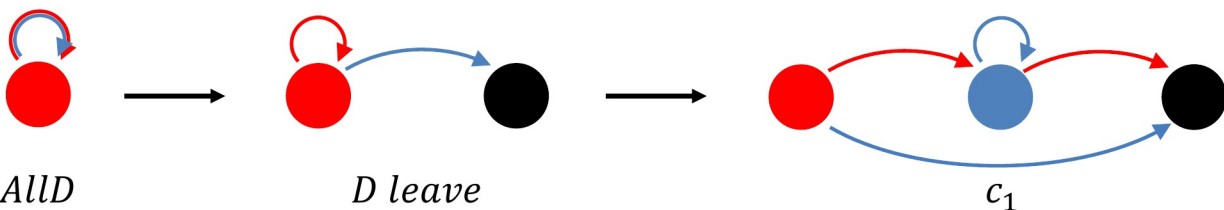

**Fig 4. An indirect invasion into AllD with the option to leave.** D-leave is a neutral mutant of AllD. If D-leave has gone to fixation, $c_1$ can invade.

## Results

### Simulations with and without the option to leave

The simulations do not have an infinitely large population. Because it is a simulation, and not a theoretical model, we can moreover not simply assume that the population always is in short-run equilibrium. However, even a moderately large finite population tends to be relatively close to a short-run equilibrium almost all of the time. More importantly, the long-run dynamics we see in the simulations match what the theory predicts, as we observe sequences of indirect invasions.

The comparison we make here is straightforward. In one set of simulations, the output in any state of an FSA can only be to cooperate or to defect. In the other set of simulations, apart from the initial state, the output in all other states can also be to leave. In the first set, without the option to leave, the model then reverts back to the standard model of repeated prisoner's dilemma, with a continuation probability that is equal to the probability $\delta$ with which pairs are not broken up exogenoulsy (see S1 Text for technical details). We then ran the simulations for a range of $b$'s (which should be interpreted as the benefit-to-cost ratio, since we normalized the $c$'s to 1), and a range of $\delta$'s, where $1 - \delta$ is the exogenous breakup probability. Comparing the average amount of cooperation with and without the option to leave, we find that *the option to leave elevates cooperation levels substantially*. For all combinations of $b$ and $\delta$, cooperation is at least as high with the option to leave as it is without, and the difference is sizable; if we take the average amount of cooperation over the parameter space without leaving—that is: for $\delta$ from 0.01 to 0.95 in steps of 0.02, and for $b$ from 1 to 6 in steps of 0.1—and compare it to the average amount with the option to leave, then the latter is 42% higher. We should obviously not attach deeper meaning to this exact number, because it is the result of a somewhat arbitrary choice to stop at $b = 6$. If we were to restrict the parameter space to benefit-to-cost ratios between 1 and 5, the difference would be larger than 42%, and if we restrict it to benefit-to-cost ratios between 1 and 7 the gap would be a bit smaller. The number does however justify summarizing the observation that, compared to panel A in Fig 5, in panel B the cooperation levels are lifted up to a substantial degree. Simulation results with alternative mutation procedures suggest that this is not an artefact of the particulars of the mutation procedure (see S1 Text).

A change from a setting without to a setting with the option to leave implies an expansion of the set of strategies. FSAs that only have states in which they cooperate or defect are obviously allowed for, both when leaving is possible, and when it is not. FSAs that also have states in which the output is that it leaves, on the other hand, are only included in the set of strategies when leaving is allowed for. The expansion of the set of strategies means that in the equilibrium analysis, there are more mutants to consider. For some strategies that are equilibria without the option to leave, that means that they stop being equilibria with the option to leave. This

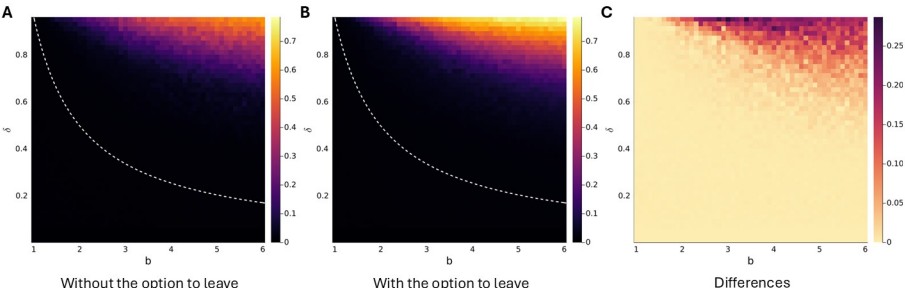

**Fig 5. Simulation results.** Panel A reflects average cooperation levels for a range of benefit-to-cost ratios $b$ and continuation probabilities $\delta$ without the option to leave. Panel B does the same, but with the option to leave. The population size is $N = 100$. Selection or mutation steps happen at a rate of 0.05 per stage game per matched pair. At a selection or a mutation step, the pair is broken up, and the strategies are replaced by offspring from strategies in the current population, in case of selection, or by a mutant. The rate at which selection events or mutations happen implies that $\delta$ has an upper bound of 0.95. In expectation, one mutation happens per 250 selection events. The color scale in panels A and B runs from 0 to 0.785, which is the highest average cooperation level in panel B. Below the dotted line, no cooperative equilibria exist, both with and without the option to leave. Panel C displays the difference in average cooperation levels between A and B.

includes strategies that start cooperating in the first round, as noted at the end of the [Introduction](). Other strategies continue to be Nash equilibria, but might nonetheless be left through indirect invasions at a higher or lower rate. On the other hand, extending the set of strategies not only means that there are more mutants to consider, but also more residents that can be equilibria.

In the following sections, we will try to identify reasons why there is more cooperation with the option to leave than there is without. All of the ingredients mentioned above will be part of the answer. In the next section we show that in the version of the game with the option to leave, there are indeed new equilibria that punish by leaving, and we will show that these equilibria are relatively stable. In Section **Getting away from AllD at a higher rate** we will point to the fact that fully defecting equilibria are invaded at higher rate if leaving is allowed for.

## Relative stability of cooperative equilibria with and without leaving

In order to see how the option to leave can add equilibria that are more stable than similar equilibria without the option to leave, we turn to an example. In this example, we compare an equilibrium strategy that punishes with defection, and one that punishes with leaving. The strategy that punishes with defection is labelled $g_1$, and it is best described as Grim Trigger preceded by a 1-period trust-building phase. The other strategy is $c_1$, which also has a 1-period trust-building phase, but responds to defection after the first period by leaving (see [Fig 6]()). If $b$ and $\delta$ are sufficiently high, both are Nash equilibria—although there is an intermediate part of the parameter space where $g_1$ is, and $c_1$ is not (yet) an equilibrium. The reason for this is that after the second round, a mutant AllD in a population where all others play $g_1$ gets a payoff of 0 until the pair is broken up exogenously, while a mutant AllD in a population where all others play $c_1$ is left after the second round, and can extract benefits from their new partner. We will return to this below. When playing against copies of themselves, both of these strategies play one round of defection, and then cooperate until the pair is broken up exogenously. The only difference is that one punishes deviations with forever defection, and the other with leaving.

Both strategies can be invaded indirectly in the same way. A strategy that is identical to $g_1$, or to $c_1$, respectively, but that loses the ability to punish would be a neutral mutant for both. Given that the only difference between $g_1$ and $c_1$ is the way they punish, such a neutral mutant

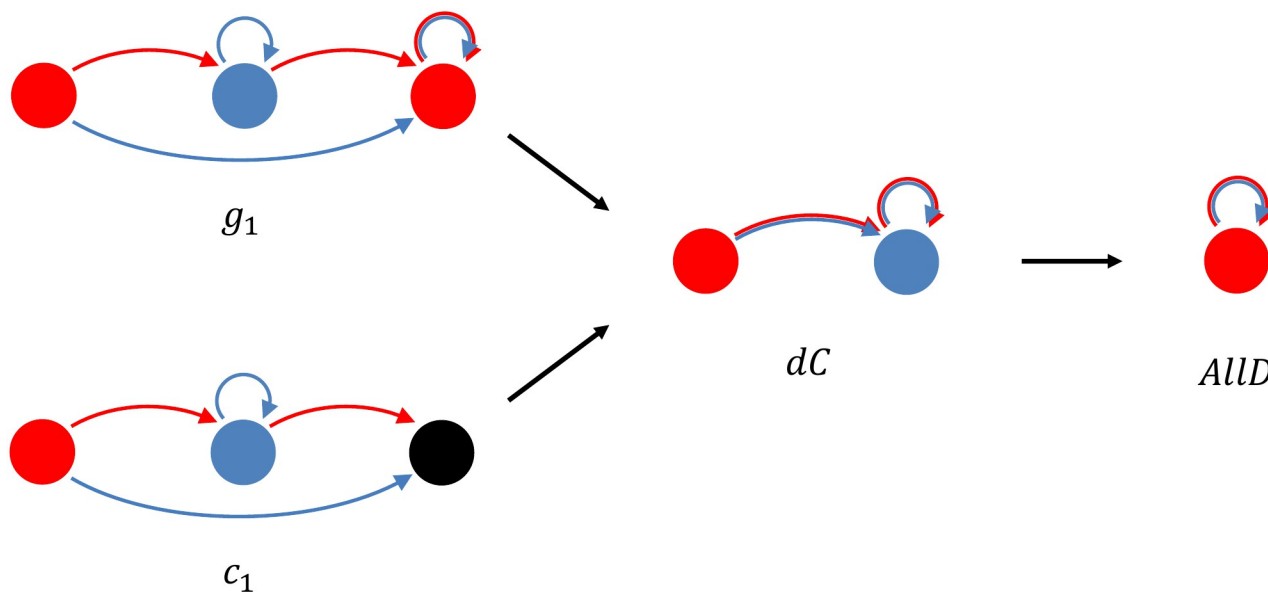

**Fig 6. Indirect invasions in $g_1$ and $c_1$.** The payoffs of $g_1$ against $g_1$, and $g_1$ against $dC$ are the same as the payoffs of $c_1$ against $c_1$, and $c_1$ against $dC$. The payoffs of $dC$ against $g_1$ and $dC$ against $c_1$ are also the same. The payoffs of AllD against $g_1$ and AllD against $c_1$ differ, but if $dC$ goes to fixation before AllD arises, then both indirect invasions are equally likely to succeed.

ends up being the exact same strategy for both; this would be $dC$ (see Fig 6). Strategy $dC$ is neutral, both for $g_1$ and for $c_1$, and therefore it has a fixation probability of $\frac{1}{N}$, where $N$ is the population size.

If we can assume that the first mutant has either gone extinct, or gone to fixation, before the next mutant appears, then these indirect invasions into either $g_1$ or $c_1$ are equally likely to succeed, because also the second step in the indirect invasion is identical.

If the mutation rate is not low enough to justify this assumption, however, there is a difference. If AllD enters the population at a point in time at which both the resident ($g_1$ or $c_1$) and the first mutant ($dC$) are still present, this takes the population to the interior of the simplex (i.e., to a mix of all three strategies). In the interior of the simplex, the replicator dynamics are different, and since the replicator dynamics are also informative about the average dynamics in finite populations, the properties of the finite population dynamics will be different too. A key ingredient for this difference is that the presence of both $c_1$ and AllD in the same population creates assortment, while with $g_1$ and AllD, this is not the case.

The easier way to see this difference is to first focus on a population of just $g_1$ and AllD, and compare it to a population of $c_1$ and AllD. With $g_1$ and AllD, all strategies just stay together, and no leaving happens in any combination. That means that we can use the standard replicator dynamics.

$$
\begin{aligned}
\dot{x}_{g_1} &= (\pi_{g_1,g_1} \cdot x_{g_1} + \pi_{g_1,AllD} \cdot x_{AllD} - \bar{\pi})x_{g_i} \\
\dot{x}_{AllD} &= (\pi_{AllD,g_1} \cdot x_{g_1} + \pi_{AllD,AllD} \cdot x_{AllD} - \bar{\pi})x_{AllD},
\end{aligned}
$$

where

$$
\begin{bmatrix} \pi_{g_1,g_1} & \pi_{g_1,AllD} \\ \pi_{AllD,g_1} & \pi_{AllD,AllD} \end{bmatrix} = (1-\delta) \begin{bmatrix} \frac{\delta}{1-\delta}(b-c) & -\delta \cdot c \\ \delta \cdot b & 0 \end{bmatrix}
$$

and

$$\bar{\pi} = \begin{bmatrix} x_{g_1} & x_{AllD} \end{bmatrix} \begin{bmatrix} \pi_{g_1,g_1} & \pi_{g_1,AllD} \\ \pi_{AllD,g_1} & \pi_{AllD,AllD} \end{bmatrix} \begin{bmatrix} x_{g_1} \\ x_{AllD} \end{bmatrix}.$$

The four constants in the payoff matrix are the average per-period payoffs for the four possible pairs, which is $(1-\delta)$ times the total discounted expected payoffs that we normally use in settings without leaving. This normalization only affects the speed of the replicator dynamics.

With $c_1$ and AllD, on the other hand, the $c_1$-players stick together, while they dissociate from the AllD players. That implies that the share of $c_1$-players that is playing with other $c_1$-players stops being linear in the total share of $c_1$-players present in the population. The average payoffs therefore are also no longer linear in the shares of the two strategies, and we need to write the replicator dynamics in a more general form.

$$\begin{aligned} \dot{x}_{c_1} &= (v_{c_1}(x_{c_1}) - \bar{v})x_{c_1} \\ \dot{x}_{AllD} &= (v_{AllD}(x_{c_1}) - \bar{v})x_{AllD}, \end{aligned}$$

where

$$\bar{v} = v_{c_1}(x_{c_1}) \cdot x_{c_1} + v_{AllD}(x_{c_1}) \cdot x_{AllD}$$

and

$$x_{AllD} = 1 - x_{c_1}.$$

S1 Text shows how average per-period payoffs $v_{c_1}(x_{c_1})$ and $v_{AllD}(x_{c_1})$ are calculated.

At really low frequencies of AllD, $g_1$ actually does a better job at suppressing AllD payoffs, because $g_1$ "binds" the mutant AllD's, and after allowing AllD to get a payoff of $b$ once, $g_1$ then holds AllD down to a payoff of 0 in all subsequent periods. Strategy $c_1$, on the other hand, cuts AllD loose, which allows it to go on and exploit other $c_1$'s. This implies that at the point of invasion (on the very left of panels C and D of Fig 7), AllD actually gets higher payoffs with $c_1$ than it does with $g_1$.

At higher frequencies of AllD, however, the assortment that $c_1$ creates by staying with other $c_1$'s, but dissociating from AllD's implies that AllD's mostly find other AllD's in the matching pool. This assortment suppresses the payoffs to those that play AllD, when the resident is $c_1$. At low frequencies, AllD payoffs therefore are suppressed more when the resident is $g_1$, while at higher frequencies, AllD payoffs are suppressed more when the resident is $c_1$. For higher $b$ and $\delta$, the latter effect overpowers the former, making $c_1$ more stable against AllD than $g_1$ is against AllD.

Comparing mixes of $g_1$ and AllD with mixes of $c_1$ and AllD helps understand why there is endogenous assortment with $c_1$, and not with $g_1$. The more relevant effect on the stability of the two equilibria we consider here, however, is due to the differences in the interior of the simplex, where three strategies are present. When the strategies present are $g_1$, $dC$, and AllD, then also in the interior of the simplex, there is no assortment. With $c_1$, $dC$, and AllD, on the other hand, there is assortment. The assortment reduces the size of the basin of attraction of AllD (see Fig 7A and 7B). Moreover, along trajectories in the interior of the simplex that are outside of the basin of attraction of AllD, selection weeds out more $dC$ in panel (b) than their counterparts in panel (a) that start at the same points.

Both of these observations are relevant if we compare the likelihood of a successful indirect invasion into $c_1$ with the likelihood of a successful indirect invasion into $g_1$ in finite population dynamics. If first a mutant $dC$ appears, random drift may take the population from the top

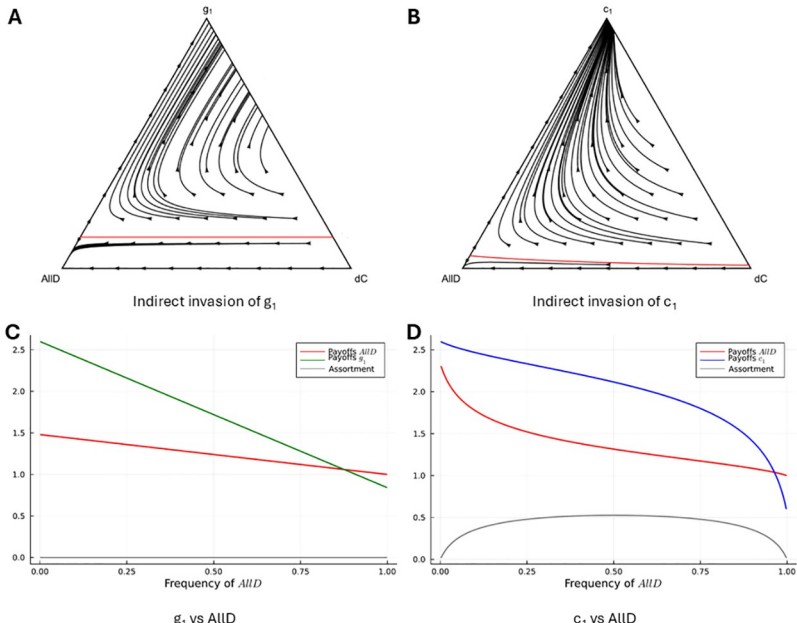

**Fig 7. Replicator dynamics for indirect invasions into $g_1$ and $c_1$.** The red lines in panels A and B delineate the basins of attraction of AllD. Besides the basin of attraction of AllD being smaller in panel B, on trajectories in the interior that do not converge to AllD, more $dC$ is weeded out along the way. Panels C and D provide details relevant to the replicator dynamics on the left edges of the simplex. Gray lines indicate assortment, or relatedness, calculated as the probability with which type $i$ individuals are matched with other type $i$ individuals minus the probability with which an individual of type $j \neq i$ is matched with type $i$. In other words, $r = P(i|i) - P(i|j)$ for $j \neq i$. Also in the interior of the simplex in panel B, but not in panel A, there will be assortment. Payoffs, assortment, and the replicator dynamics are all calculated under the assumption that the distribution of players over pair-types, and over rounds of play, is stationary [47]. The parameter values used are $b = 3$ and $\delta = 0.8$.

vertex of the simplex to states on the line segment between $g_1$, or $c_1$, and $dC$ (the right edge). If a second mutant AllD appears while the population is on this edge, it then moves to the interior. AllD subsequently is likely to take over the population, if its appearance puts the population in the basin of attraction of AllD (that is: below the red lines in the respective simplices). As Fig 7 shows, starting at the top of the simplex, the basin of attraction is easier to reach in panel A than in panel B. Therefore, in a finite population, AllD is more likely to go to fixation if the original resident is $g_1$ than it is if the original resident is $c_1$. On top of that, if the second mutant arrives at a point above the red line in panel (a)—and therefore also above the red line in panel (b)—then, starting at the same point in both panels, more $dC$ will be weeded out along the way as AllD goes extinct in panel (b) compared to panel (a). Leaving the population closer to the top vertex of the simplex makes it more likely that, subsequently, $c_1$ goes to fixation before a new mutant appears than it is that $g_1$ goes to fixation before a new mutant appears. That makes it less likely that neutral drift brings the population to a point where AllD can successfully invade when the next mutant arrives. Thus, in finite populations, overcoming the tides and currents against this indirect invasion is harder when $c_1$ is the resident, and it takes, on average, more mutations, and therefore more time, to successfully leave $c_1$.

We can also see the effect on the relative stability of $c_1$ and $g_1$ of not being in the low-mutation limit by calculating invariant distributions in a finite population for a strategy set that only consists of those four strategies. Increasing the mutation rate increases the time spent in the interior of the simplex, and it increases the incidence of new mutants arising before the

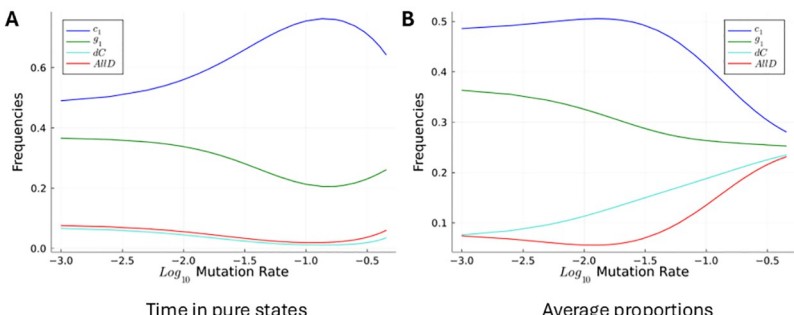

**Fig 8. Relative stability of $c_1$ and $g_1$ as a function of the mutation rate.** The plots show properties of the stationary distribution of a Moran process with parameters $N = 40$, $\delta = 0.8$ and $b = 3$ and strategies $c_1$, $g_1$, $dC$, and AllD. Strategies mutate with equal probabilities into any of the other strategies. The horizontal axis indicates the ratio of mutation steps relative to selection steps. E.g., at a mutation rate of $-1$, a mutation happens every 10 selection events, and at a mutation rate of $-2$, a mutation happens every 100 selection events. Panel A shows, out of the time that is spent at monomorphic population states, how much of it is spent at the different strategies respectively. Panel B shows the average overall frequencies of individuals of the respective types.

previous one has gone to fixation or has gone extinct. This comes with an increase in the time spent in the $c_1$ equilibrium relative to the time spent in the $g_1$ equilibrium, and it increases the average share of $c_1$ (see Fig 8). Both of these effects are only reversed when the mutation rate approaches the point where mutation becomes the only ingredient of the dynamics, leaving no room for selection.

## When punishing with leaving is better than punishing with defecting

The four-strategy model above indicates how equilibria that punish by leaving can be more resistant to indirect invasions compared to equilibria that punish by defecting, away from the low-mutation limit. Using the same set of strategies, but without $dC$, we can also see how, if leaving is an option, punishing by leaving can outperform punishing by defecting in direct competition between the two modes of punishment. If we focus on a population consisting of strategies $c_1$, $g_1$, and AllD only, then what matters for whether $c_1$ or $g_1$ performs better is the likelihood with which, after breaking up with an AllD player, one is re-matched to an AllD player. If the probability of trading in one AllD partner for another is high, it is better to be $g_1$, and sit the current match out. This will result in getting the mutual defection payoff while it lasts, but that is better than risking wasting another second-round cooperation on a new AllD player. If the probability of being matched again with yet another AllD player is not too high, on the other hand, it is better to be $c_1$ and leave, in the hope of finding a more cooperative partner. The threshold frequency for when the prospect of establishing mutual cooperation makes it worth the risk is favorable for $c_1$; only at very high frequencies of AllD is it better to punish with defection (see Fig 9). Moreover, if punishing by defecting has an advantage over punishing by leaving, both are already losing to AllD. This can be seen by the blue line being inside of the basin of attraction of AllD, the boundary of which is indicated by the red line in Fig 9. Everywhere outside the basin of attraction of AllD, and therefore all along all paths where cooperation ends up prevailing, $c_1$ always outperforms $g_1$ (see also S1 Text for a proof that this is true for all values of $b$ and $\delta$, and for $g_n$ and $c_n$ for all $n \geq 1$).

## Getting away from AllD at a higher rate

The dynamics, both with and without the option to leave, tend to go through similar phases. A population state in which there is no cooperation whatsoever is invaded indirectly, after which

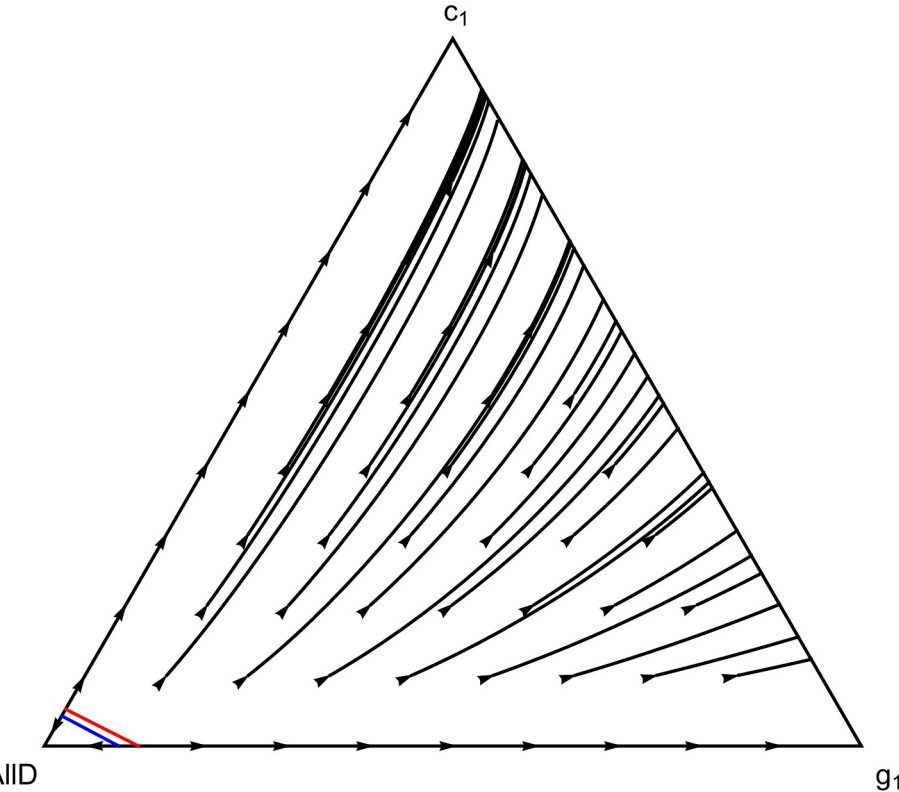

**Fig 9. Leaving or defecting.** Everywhere, except for the area down/left from the blue line, $c_1$, which punishes with leaving, outperforms $g_1$, which punishes with defection. The small area where $g_1$ outperforms $c_1$ lies entirely within the basin of attraction of AllD, which is delineated by the red line. S1 Text contains a proof that this holds for all values of $b$ and $\delta$, and also for pairs of cooperative strategies with longer trust-building phases. This implies that, all else equal, if reciprocity evolves, those that punish with leaving always do better than those that punish with defection. The parameter values used for this simplex are $b = 3$ and $\delta = 0.8$.

the population settles on a cooperative equilibrium. A subsequent indirect invasion then takes it back to a fully defecting equilibrium, such as AllD. Sometimes an indirect invasion will take the population from one equilibrium with a positive amount of cooperation to another one with a different amount of cooperation, but transitions with a complete loss of cooperation are sufficiently more frequent to ensure that the population returns to the set of fully defecting equilibria very regularly. Given that cooperation tends to break down completely, before it is re-established, both with and without the option to leave, any change in the rate at which states like AllD are left is consequential.

As is illustrated in Fig 10, the average time it takes for mutation and selection to find a path out of equilibria that are equivalent to AllD is lower in the setting with the option to leave than it is without it. This also contributes to the fact that there is more cooperation in the game with the option to leave than there is without, even though the option to leave limits the effectiveness of punishment with defection.

## Model choices

There are many ways in which our model is stylized. Below, we will go over a few model choices we made, and discuss how restricting they are. We will also discuss how different or similar they are to choices made in other papers.

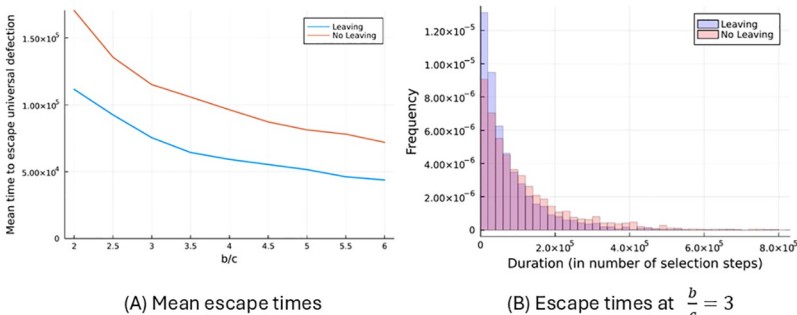

(A) Mean escape times

(B) Escape times at $\frac{b}{c} = 3$

**Fig 10. Stability of universal defection with and without the option to leave.** For a variety of $b/c$-ratios, panel A shows how much time (measured as the number of consecutive selection steps) the population spends on average with a resident that only defects, before a more cooperative mutant successfully invades. The continuation probability is fixed at $\delta = 0.9$. Panel B shows the distribution of escape times for a $b/c$-ratio of 3. The arrival rate of mutants is the same, with or without the option to leave. The sets of possible mutants and the distributions over those are different between the two settings due to the difference in feasible strategies.

First of all, our model looks at repeated prisoner's dilemma, which means that, if we add the option to leave, individuals still have the option to respond to defection with defection. That allows us to evaluate what happens if both responding in kind and responding by leaving are possible. There are however also papers that explore what leaving can do to cooperation, without comparing it to the effectiveness of reciprocity. An elegant way to do that, is to endow individuals with a level of cooperation, or generosity, and with a threshold for how much generosity they need to experience from their partner for them not to leave, which is also referred to as their choosiness. This is what happens in [57] and [27]. On all dimensions we discuss below, our paper makes the same choices as the latter study.

We wanted to integrate the option to leave in a classical repeated games setting. Some papers that do retain the classical repeated prisoner's dilemma setting choose a natural, well-argued subset of strategies [38, 39], sometimes in an Axelrod tournament style [30, 33, 37, 64]. Others, including ours, allow for a general, unrestricted strategy set [31, 34–36, 41–43, 45–50].

In our model, all individuals play every stage game with exactly one partner. This aspect is shared with most work on repeated games, but especially in the domain of partner choice, one could imagine settings in which how many interaction partners one has varies, and is the result of choices made by the individual. Theory papers that allow for a varying number of partners include [58–61]. Experimental papers that make the same choice include [51–53, 56]. The possibility to keep more than one partner by cooperating, or end up having no partner at all by defecting, might enhance the effect of partner choice. In all of the papers mentioned above, individuals are restricted to play the same stage game action (C or D) with all partners. One could also imagine realistic scenarios in which individuals differentiate between partners, depending on how they behave.

In the **Introduction**, we mentioned that in our model, when new partnerships are formed, the partners do not learn anything about past interactions that their partners had with others. We chose for individuals to not have information about interactions other than their own in order not to open the door for reputation formation or indirect reciprocity. This is a choice at one end of the spectrum, where in reality there might be some information flow. Theory papers with information flow include [33, 37, 64], empirical papers with information flow include [51–53]. More information spillover may raise the benefits of cooperating, and also enhance the effect of partner choice.

Another stylized property is that we assume what one could call a well-mixed matching pool; within the matching pool, all pairs are equally likely to be formed. In reality, there might be an exogenous, or an evolving population structure that limits the available partners, or makes some pairs more likely to be formed than others [38–40, 61].

Updating is also global in our model. If individuals in a pair die, the only thing that determines the probability with which other individuals produce offspring that replaces them, is their payoffs, and not, for instance, their proximity in a network. A network structure may however also affect who gets to reproduce where at the update event [40, 58–60].

In our stylized model, the absence of population structure, both regarding pair formation, and regarding updating, means that there is no exogenous assortment; all assortment is endogenous and only due to the choice to leave uncooperative partners. In that sense our model with leaving isolates partner choice as an ingredient.

Finally, we assume that all strategies are executed without errors. The effect of errors is investigated in [3, 19, 62]. The latter also includes a result that implies that vanishingly small error rates have a vanishingly small effect on what strategies evolve.

## Discussion

The two mechanisms that received the lion share of the attention in the literature on the evolution of cooperation are kin selection—sometimes also classified as population structure—and repetition. Population structure typically refers to any deviation from a well-mixed population, in which individuals are matched randomly. This includes interactions on networks [65–69], or within groups [70–75]. In those models, local dispersal causes neighbouring individuals, or individuals within the same group, to have an increased probability of being identical by descent, and when they do, the mechanism at work is kin selection [76–78].

Our model relates, first and foremost, to the second mechanism, in which repetition allows for reciprocity and cooperation to co-evolve [1–19]. Our version deviates from the standard setup, in that it allows for individuals to leave their current partner, and seek out someone else to play prisoner's dilemmas with. This shortens the long arm of reciprocity, because it allows individuals to run from punishment by defection. The option to leave however turns out to increase rather than reduce the average amount of cooperation that evolves. By allowing individuals to get up and leave, the model also introduces the possibility of partner choice, and this can create endogenous assortment in mixed populations. Away from the low mutation limit, this can make equilibria in which defections are punished with leaving more stable than equilibria in which defections are punished with defections. Partner choice therefore seems to be at least as powerful a mechanism for the evolution of cooperation in repeated games as reciprocity is. Already in a minimal setting, in which players have no prior information about the partner they are matched with, and only have the interactions within the repeated game itself to base their decisions to stay or to go on, this mechanism works very well.

Why humans cooperate with their siblings, even if it is costly, or why we are altruistic towards our offspring is well explained by kin selection. The research on the evolution of human cooperation therefore naturally centers around the question why we also cooperate with non-kin. For this we tend to turn towards repeated games and the reciprocity that can evolve there, or to the interaction between repetition and population structure [11, 79]. Our model points to the power of a third mechanism, which is partner choice [25–29, 57]. The assortment that this generates is different from the exogenous assortment that features in kin selection models. The assortment in our model is endogenous, and not based on identity by descent. Individuals in our model stay with their partner purely based on the experienced *phenotype* of their partner, and are not playing with others that are related to them, where

relatedness would determine the probability with which they inherited their strategy from the same individual. This phenotypic assortment, where unrelated, similarly dependable cooperators end up playing with each other, may be a better match with the long-lasting cooperation we observe in humans, who tend to exert some influence over who they cooperate with, if they can, and who cooperate with genetically unrelated others.

## Supporting information

**S1 Text.** The Supporting Information gives more detail regarding the theoretical model and the way it is simulated. It also gives theoretical results mentioned in the Main Text, with proofs. It is subdivided as follows. **The Model.** *Calculating payoffs. Why these are the average payoffs, if we assume short-run equilibrium. With and without leaving in one setting. Frequencies in the matching pool and in the population as a whole. Histories and strategies. Finite state automata.* **Simulations.** *Different mutation procedures. Algorithm for the simulations.* **Theoretical results.** *No ESS. No strategy that is RAII. Pure strategies with a trust-building phase.* (PDF)

## Author Contributions

**Conceptualization:** Christopher Graser, Matthijs van Veelen.

**Formal analysis:** Christopher Graser, Matthijs van Veelen.

**Methodology:** Christopher Graser, Takako Fujiwara-Greve, Julián García, Matthijs van Veelen.

**Software:** Christopher Graser, Julián García.

**Validation:** Christopher Graser, Matthijs van Veelen.

**Writing – original draft:** Christopher Graser, Takako Fujiwara-Greve, Julián García, Matthijs van Veelen.

**Writing – review & editing:** Christopher Graser, Takako Fujiwara-Greve, Julián García, Matthijs van Veelen.

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
