## [Decision Letter · Decision Letter 0]

6 Sep 2024

Dear Professor van Veelen,

Thank you very much for submitting your manuscript "Repeated games and partner choice" for consideration at PLOS Computational Biology.

As with all papers reviewed by the journal, your manuscript was reviewed by members of the editorial board and by several independent reviewers. In light of the reviews (below this email), we would like to invite the resubmission of a significantly-revised version that takes into account the reviewers' comments.

The paper has been evaluated by two reviewers who are experts in evolutionary game theory and repeated games. Both praise the basic setup of the paper. Based on my own reading, I fully concur: the relationship between partner choice and reciprocity is an important but understudied topic. Hence the paper certainly is, in principle, suitable for publication in PLoS Computational Biology.

However, both reviewers also make a number of extremely constructive comments to further improve the paper. I'd like to ask the authors to take these comments into account when resubmitting their revised manuscript.

We cannot make any decision about publication until we have seen the revised manuscript and your response to the reviewers' comments. Your revised manuscript is also likely to be sent to reviewers for further evaluation.

Sincerely,

Christian Hilbe

Academic Editor

PLOS Computational Biology

Feilim Mac Gabhann

Editor-in-Chief

PLOS Computational Biology

The paper has been evaluated by two reviewers who are experts in evolutionary game theory and repeated games. Both praise the basic setup of the paper. Based on my own reading, I fully concur: the relationship between partner choice and reciprocity is an important but understudied topic. Hence the paper certainly is, in principle, suitable for publication in PLoS Computational Biology.

However, both reviewers also make a number of extremely constructive comments to further improve the paper. I'd like to ask the authors to take these comments into account when resubmitting their revised manuscript.

Reviewer's Responses to Questions

**Comments to the Authors:**

Reviewer #1: This manuscript presents a novel approach to the study of repeated games by incorporating the option for individuals to leave their current partners and seek new ones. The authors argue that this option increases the overall level of cooperation in a population by creating endogenous phenotypic assortment. The model developed provides a better fit with real-world human interactions compared to standard models, where partner choice is not considered.

This approach is particularly commendable because, while the concept of an "option to leave" has been explored in previous agent-based models—such as those by Aktipis, where individuals possess mobility to change partners—this manuscript takes a step further by rigorously analyzing the impact of this option within a full game-theoretical framework. This rigorous approach allows for a deeper understanding of how the ability to leave affects the evolution of cooperation, offering a more comprehensive and theoretically grounded perspective.

Overall, this manuscript provides a rigorous analysis of the impact of the option to leave on the evolution of cooperation and is a strong candidate for publication in PLOS Computational Biology. However, there are several areas that require improvement. Once these areas are addressed, I believe this manuscript could become an excellent contribution to the field. I would be happy to re-evaluate the manuscript once these improvements have been made.

While the mathematical analysis in this manuscript is reliable, it may pose some challenges for readers of PLOS Computational Biology. Therefore, I recommend that the revised manuscript focus on improving accessibility in this area. Below are several suggestions for improvement.

For example, the equilibrium analysis in Section 3 might be more readily understood by readers familiar with economic game theory, such as those who regularly engage with journals like the Journal of Economic Theory or Games and Economic Behavior. However, for those in fields like mathematical biology or computational biology, the text-based explanation alone might be difficult to grasp (I also found Section 3 challenging to fully understand). To aid in comprehension, I suggest including graphical illustrations that represent the logic of Section 3. Specifically, using finite state automata to illustrate how indirect invasions progress in both scenarios—with and without the option to leave—could significantly help readers understand the underlying logic.

On the other hand, the content of Sections 4 and 5 is more accessible to those in mathematical biology (I myself gained a clearer understanding of the paper’s implications after reading these sections). However, these sections also have shortcomings. Although I am familiar with this type of simulation, I found Section 4 insufficiently detailed; even after consulting the supplemental material, I would not be able to replicate the simulations described. I suggest including "pseudocode" for the simulations in Section 4. This practice is common in this journal and would greatly enhance reproducibility.

Section 5 also appears to lack sufficient information. At the very least, the replicator equations should be provided. Without these equations, it is difficult to envision the evolutionary dynamics that the authors describe.

Beyond the analytical aspects, I would like to highlight a different concern. In reality, human interactions are often network-based, meaning that individuals typically make partner choices within the constraints of their social networks rather than within an infinitely large population as assumed in the model. For example, Zimmermann and Eguíluz, Physical Review E, (2005) demonstrated through an agent-based model in a social network that cooperation can evolve when agents sever ties with defectors. This mechanism is similar to the one presented in this manuscript, where the option to leave reduces the payoff of All D strategies. It would be beneficial to discuss existing research on the option to leave in the context of social networks, and I suggest incorporating this discussion into the existing section on social networks in the Discussion. This would provide a more comprehensive view and acknowledge the importance of network structure in real-world cooperation.

Minor Comments:

Lastly, there are issues with the order in which figures are referenced and their placement within the manuscript. For instance, it is problematic that Figure 2 is referenced before Figure 1 in the text, and that the results from the simulations in Figure 1 are presented before the corresponding discussion in Section 4. I recommend revising the manuscript to ensure that figures are introduced in the order they are discussed, and that they appear in proximity to the relevant text to enhance readability and coherence.

Reviewer #2: In "Repeated games and partner choice," Graser et al develop and analyze a model of repeated PD play with partner choice based only on the option to end one's current relationship. They compare this model to repeated PD play without the exit option and thus without endogenous partner choice. They do not consider partner choice without repeated PD play because their partner choice mechanism, which is quite compelling for reasons explained below, only works when repeated play is possible. They find that adding the exit option to repeated play supports the evolution of cooperation better than repeated interactions alone over a substantial region of parameter space.

This research is important for at least four fundamental reasons.

(1) It analyzes what happens when two mechanisms combine either to support the evolution of cooperation or not. We do have other examples of work combining mechanisms (e.g. Axelrod and Hamilton, 1981; van Veelen et al, 2012). But, it's also true that a bizarre acrimony has burdened work on the evolution of cooperation for a long time. This acrimony has little to do with science and almost everything to do with different tribes of researchers grinding their respective axes. This kind of thing happens. Why it happens in one field more than another? Who knows? Whatever the reason, the acrimony related to cooperation, I suspect, has meant that approaches combining mechanisms have received less attention than they would have in a counterfactual world without the acrimony. The current paper reminds us that, when it comes to the evolution of cooperation, hypothesized mechanisms are rarely incompatible. It also highlights how to move beyond the dysfunctional divisions that have marred the field in the past.

(2) Less generically, the current paper presents an especially convincing form of partner choice. It's straightforward and intuitive to argue that individuals probably do not pair off randomly, and this remains true even once we account for any biases in matching due to population structure. This raises the possibility that partner choice, however it works, creates some extra measure of assortment, and this is why it's interesting in relation to cooperation. That said, partner choice mechanisms can vary widely in terms of their informational assumptions. At one extreme, assume that everyone can directly and accurately observe the choices of everyone else. In such a world, everyone has a lot of reliable information about everyone. It's easy to imagine adding a partner choice mechanism that would allow cooperative individuals to identify each other and pair off endogenously. But, the amount of information in the system and the value of the information in the system are exogenously assured in a way that strains belief.

The current paper presents a partner choice mechanism at the opposite extreme. All individuals know is whether they like their current partners or not. Anyone who does not can end the current relationship and enter a pool of unmatched players available for random rematching. This pool is not representative of the entire population, and this bias is what creates the informational regularity that gives partner choice its force. In this case, the model does not strain belief at all. The informational requirements from the perspective of the individual are minimal and completely reasonable. All I have to know is whether I like the guy I'm currently interacting with.

(3) With the use of FSAs to encode strategies for repeated play, the current model effectively allows an infinitely large strategy space. This is important because it increases confidence that any results are not artifacts of assumptions the researchers have made about the strategy space. The classic (and simplest) example of why assumptions about the strategy space matter, of course, is repeated PD with {ALLD,TFT} as a strategy space versus {ALLD,TFT,ALLC}. Other research (e.g. van Veelen et al, 2012) has suggested that groundless limits on the strategy space may be a general concern. But, the use of FSAs removes this worry, and this is another virtue of the paper.

(4) The paper opens with a very nice discussion of how the option to leave one's current partner has two countervailing effects on the evolution of cooperation. This raises the question, which effect is strongest and when? I found this framing to be quite compelling.

With these positives established, I would also like to offer several comments, questions, and suggestions. I begin with two suggested extensions for the paper. In my view, these extensions would provide a more complete picture of when and how repeated interactions and partner choice combine to support cooperation or not. I follow with a number of suggestions, comments, and questions about the writing. The paper varies a lot in terms of how clear and accessible different passages are. Some passages do a really great job of explaining ideas in an intuitive, precise, and accessible way. Other passages can be quite difficult to understand, and in some cases I'm simply not sure what the authors want to say. I flag the difficult passages so the authors can consider other ways of explaining whatever it is they want to explain. I offer specific suggestions when I have them.

Extensions:

(A) Include implementation errors in the simulations. Presumably, results remain similar for sufficiently low error rates, but where is the boundary? What happens as the amount of noise in the translation from strategies to behavior increases? I doubt the value of the paper can go down depending on the answers to these questions, but it can definitely go up. It's important to know the scope of the interaction between repeated interactions and partner choice, and varying the rate of implementation errors would almost certainly help the reader evaluate this scope.

(B) In the supplement, the authors describe - briefly - the different mutational algorithms they used. The supplement should include more detail. I'm not suggesting additional analyses, just more information about the analyses the authors have apparently already done. First, the description of the different algorithms (e.g. bottom of p. 12) in the supplement is not sufficient; I really can't tell what the authors did from the all-too-brief descriptions. Second, the authors say that qualitative results remain the same regardless of the mutational algorithm used. It would be useful to include some of the other results in the supplement for readers to see this for themselves. Again, this will simply help readers evaluate the scope of how partner choice and repeated interactions combine.

The writing:

(i) On p. 3, the authors speak of reciprocity and cooperation evolving together. For my taste, this is a strange phrasing as it suggests that both reciprocity and cooperation are genetically encoded, under selection, and increasing in frequency. What actually happens is that reciprocal strategies are encoded and increasing in frequency, with cooperation as the behavioral result. Probably a matter of taste, but that was my reaction.

(ii) On p. 5, it seems quite strange to call the initial string of defections the "trust-building" phase. The citations suggest that others have already established this terminology. I don't know. But, it's a strange label given we're talking about starting off a relationship with a bunch of defections. Consider a different label or a brief explanation of the logic behind the term "trust-building" in this context.

(iii) On p. 5, the authors say they "translate" a result from a 2009 paper in the Rev of Econ Studies. What does this mean?

(iv) On p. 5, the authors provide a nice intuition about how the option to leave undermines the evolution of cooperation, but they decline to present an intuition about the upside until after they have described their model. I found this inconsistency a bit jarring. If they can provide an intuition about the downside before presenting the model, I guess they can provide an intuition about the upside.

(v) The second half of p. 7 is confusing, and I didn't really understand the point the authors want to make.

(vi) On the bottom of p. 8, the authors say that when a population is in equilibrium, punishment does not occur. This does not fit with the rest of the paragraph, which clearly suggests equilibria in which cooperation levels are positive, but maybe not as high as possible. Please clarify. Also, sentences of the following sort are often not very helpful: "The reason is similar to the reason without leaving." It's better to use more words, be less vague, and remove ambiguity.

(vii) Similarly, the top of p. 9 talks about a neutral mutant attaining a "high enough" frequency, but this is less restrictive than the mention of "fixation" in the preceding paragraph. Please clarify.

(viii) I found most of p. 9 to be unclear and challenging.

(ix) I found the emphasis on b at the bottom of p. 11 to be a little strange. Looking at Fig. 1, variation in delta seems at least as important as variation in b, if not more important. Maybe the authors include this discussion simply because b, unlike delta, has no natural upper bound. I'm not sure. I just provide my reaction for the authors to consider.

(x) Relative to Fig. 1, which provides more or less complete information, I didn't see a lot of added value for the exercise (pp. 11-12) in which the authors average over parameter space.

(xi) On p. 12, the authors say that the combination of mechanisms lifts cooperation levels "across the board" relative to repeated interactions alone. Fig. 1c does not really support this claim because, for a subset of parameter space, the difference in cooperation levels is approximately zero. In any case, I don't see a need to say something like this. It's enough to show that cooperation levels increase for the complementary subset of parameter space. Moreover, it's interesting that the big effects occur when delta levels are high, a region of parameter space in which naive intuition might suggest that repeated interactions can support high levels of cooperation without any additional mechanism.

(xii) I found the middle of p. 12 confusing.

(xiii) p. 14, "In the interior . . . population dynamics will be different too." This passage is quite vague.

(xiv) I found the analysis of escape times in the supplement very informative, and the authors might consider putting some of those results in the main text. I know space can be tight, but for me that section of the supplement was extremely useful.

(xv) The authors might consider a general, perhaps speculative discussion about when expanding the strategy space hinders versus helps the evolution of cooperation. After reading a paper like van Veelen et al (2012), one might think it only hinders. But this paper offers an example of it helping.

(xvi) The title could be more informative.

**Have the authors made all data and (if applicable) computational code underlying the findings in their manuscript fully available?**

Reviewer #1: **No: **The pseudocode for the simulation is not provided.

Reviewer #2: Yes

PLOS authors have the option to publish the peer review history of their article (what does this mean?). If published, this will include your full peer review and any attached files.

Reviewer #1: No

Reviewer #2: No
---

## [Decision Letter · Decision Letter 1]

20 Jan 2025

Dear Professor van Veelen,

We are pleased to inform you that your manuscript 'Repeated games and partner choice' has been provisionally accepted for publication in PLOS Computational Biology.

Best regards,

Christian Hilbe

Academic Editor

PLOS Computational Biology

Feilim Mac Gabhann

Editor-in-Chief

PLOS Computational Biology

In the meantime, the paper has been re-evaluated by the same two reviewers who have already handled the original submission. Both reviewers are fully satisfied with the changes and recommend acceptance. So do I.

The model is very insightful and I am sure it will have a good impact on the direct reciprocity literature.

Please address the remaining reviewer comments when submitting the final version of your paper. 

Reviewer's Responses to Questions

**Comments to the Authors:**

Reviewer #1: All of my concerns have been addressed in the revision.

However, I would like to point out that the title of section B.2 in the SI should be "Algorithm for the simulations" rather than "Pseudo-code for the simulations."

Reviewer #2: I've had the pleasure of reading Graser et al's revision, "Repeated games with partner choice," submitted to PLoS Computational Biology. This is important research. In addition, the revision reads extremely well now. My comments will be brief.

1) I suggested adding implementation errors before. I was not aware of the recent working paper on exactly this topic by Graser and van Veelen. The added discussion of errors with the citation in the revision is sufficient.

2) The discussion of mutations in the supplement is now fine.

Just a couple small additional comments.

a) On p. 13, to avoid any possible confusion, I'd recommend, "For low $\\delta$, equilibria without any cooperation therefore are less stable with the option to leave than they are without the option to leave."

b) On p. 14, the comparison is straightforward, i.e. no need for "rather."

c) On p. 29, given the taxonomy the authors have set up, Efferson et al (2016) is an example of an empirical study without information flow. Players did not know anything explicit about the past choices of new partners.

**Have the authors made all data and (if applicable) computational code underlying the findings in their manuscript fully available?**

Reviewer #1: Yes

Reviewer #2: Yes

PLOS authors have the option to publish the peer review history of their article (what does this mean?). If published, this will include your full peer review and any attached files.

Reviewer #1: No

Reviewer #2: No

---

## [Editor Report · Acceptance letter]

26 Jan 2025

PCOMPBIOL-D-24-01201R1 

Repeated games and partner choice

Dear Dr van Veelen,

I am pleased to inform you that your manuscript has been formally accepted for publication in PLOS Computational Biology. Your manuscript is now with our production department and you will be notified of the publication date in due course.

With kind regards,

Zsofia Freund
